# The influence of the solid to plasma phase transition on the generation of plasma instabilities

E. Kaselouris[1,2], V. Dimitriou[1], I. Fitilis[1], A. Skoulakis [1], G. Koundourakis[1], E.L. Clark[1], M. Bakarezos[1], I.K. Nikolos[2], N.A. Papadogiannis[1] & M. Tatarakis[1]

The study of plasma instabilities is a research topic with fundamental importance since for the majority of plasma applications they are unwanted and there is always the need for their suppression. The initiating physical processes that seed the generation of plasma instabilities are not well understood in all plasma geometries and initial states of matter. For most plasma instability studies, using linear or even nonlinear magnetohydrodynamics (MHD) theory, the most crucial step is to correctly choose the initial perturbations imposed either by a predefined perturbation, usually sinusoidal, or by randomly seed perturbations as initial conditions. Here, we demonstrate that the efficient study of the seeding mechanisms of plasma instabilities requires the incorporation of the intrinsic real physical characteristics of the solid target in an electro-thermo-mechanical multiphysics study. The present proof-of-principle study offers a perspective to the understanding of the seeding physical mechanisms in the generation of plasma instabilities.

[1] Centre for Plasma Physics & Lasers - CPPL, Technological Educational Institute of Crete, School of Applied Sciences, Rethymnon & Chania GR-73133& GR-73133, Greece. [2] School of Production Engineering & Management, Technical University of Crete, Chania GR-73100, Greece. Correspondence and requests for materials should be addressed to M.T. (email: m.tatarakis@chania.teicrete.gr)

Over the last 15 years or so there is a large interest on the seed physical mechanisms for the generation of plasma instabilities[1–6]. This fundamental interest is not only triggered by basic physics curiosity to understand plasma instabilities but also stems from important plasma applications, among others in inertial confinement fusion (ICF), either in magnetized pulsed power liners or in ICF capsule targets where strong currents or laser pulses are used to heat the target, respectively[7–10]. Plasma instabilities[2–6, 11, 12] are the result of any disturbance that may occur in a plasma parameter such as density, temperature, magnetic or electric field, or current. They dictate the plasma behaviour in most plasma geometries and plasma regimes. The suppression of plasma instabilities is difficult due to the complexity of the physical processes in plasmas. For the better understanding of the plasma instabilities, two methods of analysis are typically followed that consider the plasma either as a hydrodynamic magnetized fluid that is governed by the fluid equations, or as a statistical system determined by the kinetic equations and the appropriate distribution functions[11].

The choice of method depends on the initial phase of the target (i.e. solid, liquid or gas), the initial target geometry as well as the way energy is deposited onto the target, for example by laser–matter interaction, Ohmic heating by strong currents or by particle beam interaction. Irrespective of the approach used to study plasma instabilities, the history of the plasma generation is generally not taken into account. Usually the information of the physical processes during the phase change from solid, liquid or gas to plasma is neglected[6,7,9,13]. In addition, the use of assumptions on the plasma conditions after surface plasma initiation deteriorates the efforts towards the necessity to explore methods for mitigating the plasma instabilities[10]. On the other hand, it is clear and widely recognized by the scientific community[6,9,13,14] that the complexity of such a multiphysics problem poses serious difficulty to successfully incorporate the realistic physical initial conditions of the targets into the simulations. In other words, studying the plasma by taking into account the initial state memory of the interaction, introduces enormous complexity since it involves simultaneously occurring multi-disciplinary physical phenomena. Transient and rapidly evolving electromagnetic (EM), electro-thermo-mechanical (ETM) and magnetohydrodynamics (MHD) effects simultaneously characterize the plasma generation and therefore the onset of the plasma instabilities. Traditionally, the experimental investigation of the plasma instabilities has also focused on the plasma phase due to the difficulty of evaluating the whole-time history from solid to plasma formation[6,15,16].

These uncertainties have been the trigger of dedicated investigation of the processes related to the initial target physical conditions[10,17,18]. Until now all these studies were limited to the coupling of properties of the solid, among others the resistivity, the heat conduction, the magnetic diffusion, the viscous damping and the stress tensors, into the system of the MHD equations. These studies treat the target at its initial condition as a MHD fluid having a solid density, including terms related to some of the physical properties of the solid material. Our recent studies in solid targets suggest that the intrinsic real parameters of the target before plasma generation regimes, could critically influence the target dynamics in space and time[19,20].

In the light of the above, here we offer a perspective to these efforts, since we do not start from a MHD plasma state at solid density but from a real solid, incorporating the initial material physical properties of the target in an ETM multiphysics study of simulations and experiments. This is a major step that alleviates some of the aforementioned difficulties and provides perspectives in the study of seeding mechanisms for the generation of plasma instabilities. It must also be mentioned here, that this proof-of-principle study is not aiming to discuss in detail aspects of plasma instabilities such as the kind of instability, the growth rates, the wavelengths etc., but to present the physics that needs to be considered at the very first moments within the Joule heating. The inclusion of results for the early plasma dynamics only serves the necessity to demonstrate the validity of the physics methodology presented via the comparison with experimental data in the plasma phase of the interaction. It is found that only when the real thermo-elasto-plastic properties of the metallic material are used as a seed in the multiphysics simulation, the results match the experimental data. Any other seeding such as sinusoidal or random perturbations show different instability structure at a later time that significantly deviates from the experiments. This unambiguously demonstrates the critical role of the use of the real intrinsic physical properties of the target when the seeding mechanisms of plasma instabilities are studied.

## Results

**Experimental and numerical simulation approach.** A low, slow rise time current coupled to thick exploding metallic wires is considered in a Z-pinch geometry. Time zero is taken to be the moment of the application of the current onto the solid target. The Z-pinch target configuration is selected because it has been extensively studied with well understood plasma dynamics and known plasma instabilities[6,15]. Thick exploding wires are chosen as loaded targets since the low, slowly rising heating current conveniently allows the study of plasma generation at the skin effect mode, where all phases of matter can exist simultaneously for relatively long times. Copper (Cu) is the material of choice here since it has well known physical properties, established strength material behaviour, its equation of state (EOS) is experimentally validated[21] and has been widely used for the study of the development of instabilities in the skin effect mode[10].

For the experimental study of the dynamics of the wire target before the initialization of plasma formation, we developed a modified Fraunhofer diffraction diagnostic offering a spatial resolution of ~1 μm and therefore high accuracy in measuring the wire expansion. This diagnostic is used in conjunction with laser probing knife-edge schlieren imaging, shadowgraphy and interferometry to record the early history of the plasma formation[15,22,23] and to reliably determine when the vaporization threshold is reached and the gas phase occurs. Supplementary details are described in Supplementary Note 2.

Multiphysics numerical simulation of the dynamic response of the target during its heating and conversion into plasma has been developed based on coupled finite element (FE) and MHD methods. In order to study the wire's response to the heating source (Joule heating), a strength material model along with evaluated EOS data are used. The EOS data are necessary to describe the hydrodynamic response of the material, while the strength material model describes the deviatoric stress behaviour or distortion of the material. The density distribution and the wire radius at the last time step of the solution provided by the FEM simulation are coupled to a resistive MHD finite difference/volume code. A fluid and a plasma region, surrounded by a vacuum region, are considered for the MHD transient analysis. This choice is based on our experimental results suggesting that the plasma of the Z-pinch target consists of a dense fluid that surrounds the solid, which persists for a long time during the current discharge, surrounded by a low density hot coronal plasma. Similar plasma density distributions are observed in previous Z-pinch studies[6,13,15,22] using different Z-pinch plasma devices. The detailed coupling between the MHD and FEM analysis is described in Supplementary Note 1.

**Demonstration of results.** Figure 1 presents snapshots that compose the full picture of the temporal evolution of the heated target starting from the thermoelastic regime of the interaction up to the plasma regime. In particular, Fig. 1a–d show the simulation results of density, temperature and Von Mises stresses taken at a cross-section of the wire at various times before plasma

formation. Figure 1e–h illustrates simulation results of the temporal evolution of the instabilities from the solid to the plasma phase along the wire's half-length. As time progresses and the state of the copper wire changes, the generated instabilities in the thermoelastic regime are enhanced in amplitude but retain the same wavelength until the plasma phase starts. The density

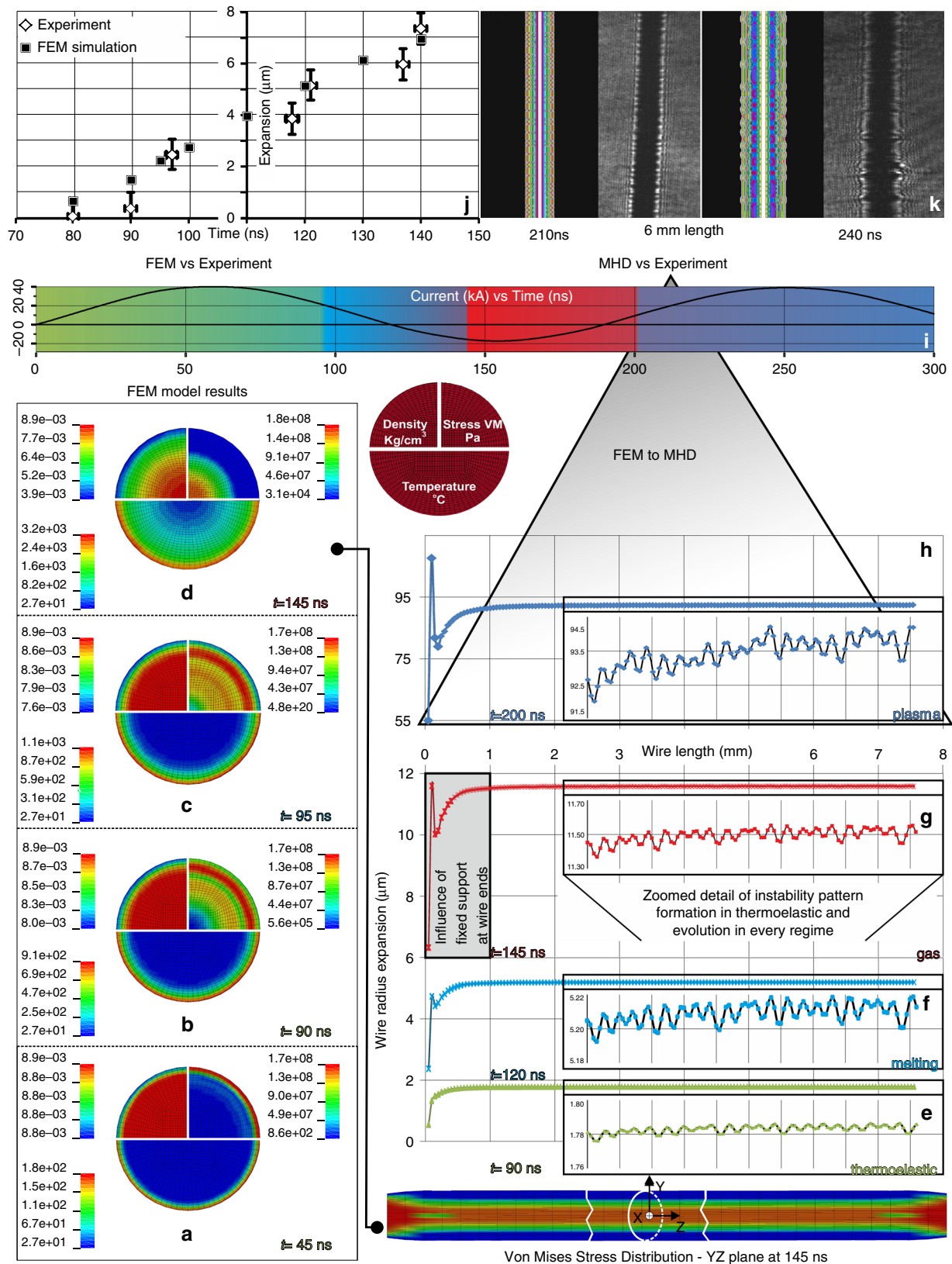

distribution given by the FEM simulation when plasma phase starts is the initial condition of the MHD code. This loop process from the FEM to the MHD is repeated until 1.5 periods of the current (300 ns for our plasma device, Fig. 1i) where plasma instabilities have been fully developed. Simulation results with a random target roughness up to 100 nm show no influence in the resulting instabilities. Other recent studies also suggest that at this small-scale roughness other physical mechanisms are the significant seed of the instabilities[7]. The FEM vs. Experiment plot in Fig. 1j presents the experimental measurements from the modified Fraunhofer diffraction laser probing diagnostic, as well as the simulation results of the radial expansion of the target until the vaporization threshold. The diameter of the metallic wire is expanded by ~2.5% due to Joule heating before it enters the gas phase regime. The good agreement validates the developed multiphysics model.

Figure 2 shows a zoomed detail of the MHD vs. Experiment snapshots of Fig. 1k for the comparison of an indicative MHD simulation with shadowgraphic experimental results. Shadowgraphy[15] offers an excellent qualitative visualization diagnostic for the plasma density distribution via the modification of the laser pulse light intensity due to the transverse plasma density gradients. There is excellent agreement of the whole

spatiotemporal dynamics of the observed plasma evolution with our physical model. In Supplementary Note 3, results are shown where the MHD study—uncoupled from the thermo-elasto-plastic phase of the target—is initiated by an artificial externally seeded perturbation function either sinusoidal, random or multispectrum periodic (Supplementary Figs. 9a and 10a). These results are compared with those obtained when the real thermo-elasto-plastic seed is used as initial perturbation. When the real thermo-elasto-plastic physics of the solid target is used as a seed the simulation results match the experimental data. For any other artificial perturbation different instability structure is observed which deviates significantly from the experiments (Supplementary Figs. 9b–g and 10b–g). This clearly and unambiguously demonstrates how crucial is to consider the real intrinsic physical properties of the target when studying the seeding mechanisms of plasma instabilities.

From the results presented in Figs. 1 and 2 the physical processes that occur during the thermo-elasto-plastic regime of the interaction and act as seeding for the initiation of plasma instabilities are explained. The imposed boundary conditions in synergy with the Joule heating of the current-carrying surface, the electromagnetic force and the resulting compressive stresses are responsible for the rapid creation of radial and longitudinal

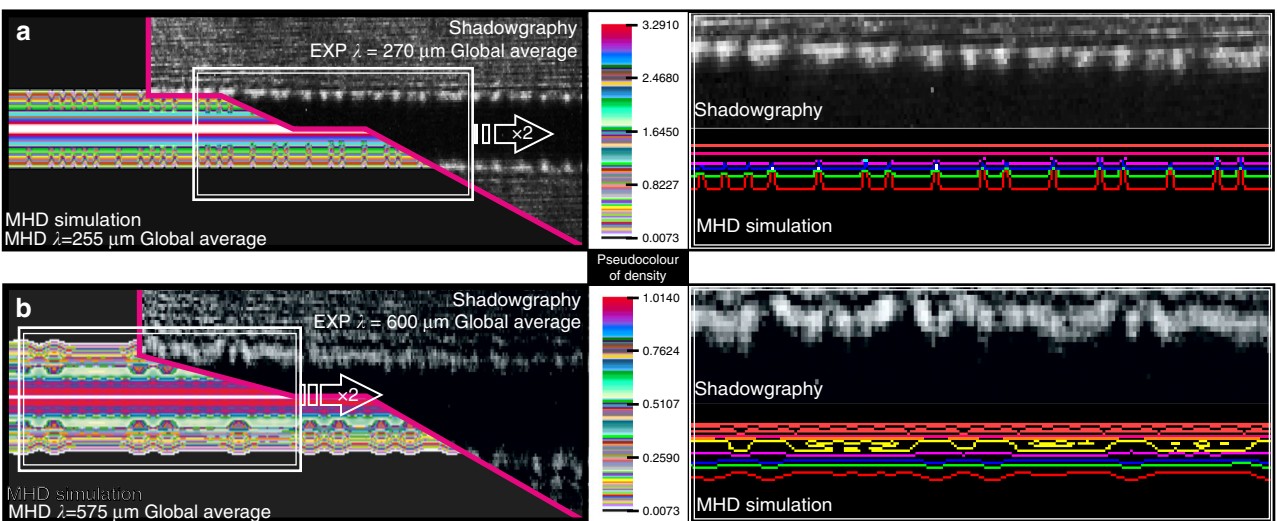

**Fig. 2** Experimental vs. magnetohydrodynamic simulation results in the plasma phase. The wavelength of the perturbations of the coronal plasma increases with time. The dominant average axial wavelength λ of instabilities in the coronal plasma (**a**) 210 ns after the current start is measured to be 270 μm and the simulation computes it to be 255 μm, while 30 ns later (**b**) is measured to be 600 μm and the simulation computes it to be 575 μm. The pseudocolour bar corresponds to mass density in arbitrary units

**Fig. 1** Time history of the wire's explosion from the thermoelastic to the plasma regime. Finite element method (FEM) cross-section *XY* plane results of density, temperature and Von Mises (VM) stress from 0 to 200 ns at regimes: **a, b** thermoelastic, **c** melting and **d** gas. Up to 92 ns the target dynamics lies in the thermoelastic regime since the maximum temperature is below the melting point of Cu (1085 °C). After 1 ns, the outer part of the target reaches a temperature above the melting point indicating the start of the liquid phase (melting). After 141 ns the outer part of the wire reaches its boiling temperature (2560 °C) indicating gas phase initiation. Coronal plasma is generated at the outer part of the wire at 200 ns as the temperature has reached the plasma temperature threshold for Cu (6000 °C)[24]. In **a**, the maximum VM stresses are located at the outer part of the wire. As time progresses, compressive elastic stresses are generated directed towards the core of the target. In **d**, stresses are higher in the core of the target, while they are lower in the outer part where the solid starts to behave like a fluid loosing its elastic properties. This is due to the increased temperatures above the melting point. Moreover, the maximum density in the core is slightly higher than the initial solid density due to the compression forces that act upon it: the compressive stresses because of inertia that oppose thermal expansion and the Lorentz force. The density of the outer part of the wire decreases in time from **a**–**d**. VM stress distribution for a longitudinal cross-section of the wire at 145 ns is also illustrated, where stresses are higher in the central part of the wire due to compression. **e**–**h** Temporal evolution of the instabilities from solid to plasma phase along the wire's half-length. Magnetohydrodynamic (MHD) simulation is initialized by FEM and loop loads the plasma density distributions. **i** Current waveform. **j** Expansion of wire until the gas phase regime starts (140 ns). Error bars indicate the measurement accuracy (±1 μm). **k** Comparison of MHD simulation with experimental picosecond laser optical probing shadowgraphy

matter oscillations along the length of the wire in the thermoelastic regime. The metallic material tends to expand, but the fixed ends do not allow the stress to relax. In the longitudinal z-direction, the fixed ends provide reaction forces due to inertia that keep the wire at its initial length. These reaction forces result in a net compressive stress[25] on the wire in the longitudinal direction. It is observed that the wire radius varies along its length, thickening in parts and thinning in others stimulating small radial surface perturbations, while significantly deforming at its ends (see influence of fixed support at wire ends in Fig. 1). The above reaction forces are instantly imposed along the whole length of the wire. The behaviour is predominantly due to the longitudinal thermal shock (due to Joule heating), which attempts to alternately shorten and lengthen the wire. This, in addition to the radial thermal shock, accounts for the change in wire's radius along its length. The time duration of the aforementioned observed phenomena eliminates the possibility that these have been generated by propagation of mechanical wave perturbations in the target. Similar target behaviour has been observed in studies aiming to test the mechanical strength of metallic wires under the influence of pulsed currents[26].

## Discussion

The agreement between our physical model and the experiment unambiguously suggests that, the pre-plasma solid target dynamics is the most important phase for the seeding mechanisms of plasma instabilities. How would the instabilities studied here evolve into the seed of a global MHD mode at a later time in the plasma phase remains an open question and requires further study.

Our coupled FEM/MHD model may be further applied to other solid target/plasma geometries and heating methods[19,20,27,28]. Phase transitions may take place at different times, if the heating source delivers different current shapes and/or magnitudes (i.e. different plasma devices) but the governing physics will remain the same as suggested by our simulations. Adaptation of our approach to known instabilities, such as the electrothermal instability (ETI) in the linear or nonlinear magnetic diffusion regime[6,7,10,14,17] that begins to grow immediately after the heating source is applied onto the solid target, will provide significant insight and is already being pursued (Supplementary Note 4). Indeed, early stage results[29] show that the inclusion of the material's physical properties modifies the ETI into an instability, which for future reference we choose to call it ETM instability, having growth rates at least one order of magnitude larger than those found in the literature. Moreover, the ETM instability could act as seeding mechanism for the outstanding helical instability structures observed in premagnetized liner experiments[30–35]. The further study of the ETM instability can enlighten the efforts for the understanding of the seeding mechanisms of such helical structures.

## Methods

**Experiments.** The Z-pinch pulsed power device is powered by a Marx-bank with a 600 J energy capacity. The Marx-bank is coupled to a water-filled pulse forming line (PFL) and a self-breaking $SF_6$ switch. This arrangement produces a peak current of 40 kA within a rise time of 60 ns. A copper wire of 300 μm diameter and 15 mm length is soldered to conical shaped electrodes placed in a chamber under a $10^{-4}$ mbar vacuum. A V-dot probe measures the derivative of the voltage at the PFL and a Rogowski groove measures the derivative of the current passing through the wire. For the optical probing diagnostics during the wire's explosion a pulsed laser of 150 ps duration is used. Charge coupled device (CCD) cameras are used for capturing the laser probing diagnostic images. In order to investigate the wire diameter dynamics before plasma formation a modified Fraunhofer diffraction method was developed. The fringe pattern is recorded at the focus distance of a lens at a time before the explosion and at a defined time after the current start. The focal spot of the laser probe beam was shifted just out of the CCD camera's frame in order to reveal the second and higher order fringes at the image. The line out of the image along the diffraction axis is used for measuring the wire's diameter and therefore the expansion of the wire as a function of time.

For the schlieren imaging a knife-edge, oriented parallel to the wire, is used at the focal length of the imaging lens and the formation of coronal plasma is revealed by the bright light that appears on the same side of the wire image as that where the knife is placed. Light deviation caused by a plasma dominated area can therefore be distinguished between that and what is produced by a neutral gas area[23]. A Mach–Zehnder interferometer in finite-fringe mode was also developed and implemented for plasma density measurements.

**Simulations.** A 3D multiphysics coupled electromagnetic thermo-structural hydrodynamic simulation based on FEM is developed using LS-DYNA® code[36,37]. The electromagnetic fields are computed by solving Maxwell equations using FEM for the conductors coupled with a boundary element method for the surrounding vacuum. Furthermore, the skin depth effect is also taken into account. When the electromagnetic fields have been computed, the Lorentz force $\mathbf{F} = \mathbf{j} \times \mathbf{B}$, where $\mathbf{j}$ is the current density and $\mathbf{B}$ the magnetic field, is evaluated at the nodes and added to the mechanical solver, which computes the deformation of the wire. The new geometry is used to compute the evolution of the EM fields in a Lagrangian way. In particular, the Joule heating power term ($j^2/\sigma$, where $\sigma$ the electrical conductivity) is added to the thermal solver in order to update the temperature. The current waveform, measured during the experiments, is used to drive the expansion of the metal wire, while the ends of the wire are at 27 °C. Phase change transitions from solid to liquid, liquid to gas and gas to plasma are taken into account based on temperature criteria. A combination of the analytical Gruneisen and tabular multiphase EOS[21] for the hydrodynamic behaviour of the copper are used together with a Johnson–Cook[38] strength material model for the study of the elastoplastic effects. The electrical conductivity vs. temperature and density is computed using Burgess[37] EOS. Temperature dependent properties of thermal expansion, thermal conductivity and specific heat, as well as the latent heat of melting are also taken into account.

The density and temperature distributions from the LS-DYNA multiphysics analysis are coupled at every time step as initial conditions to the MHD code PLUTO where an ideal gas equation of state is considered as well as Spitzer resistivity. PLUTO solves the three conservation laws of continuity, momentum, energy, the Faraday's law and the EOS[39].

**Data availability.** The data that support the findings of this study are available from the corresponding author upon request.

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

## Acknowledgements

We acknowledge financial support through the Actions: 'National Research Infrastructure for HiPER' MIS 376841 and 'ELI-LASERLAB Europe Synergy, HiPER and IPERION-CH.gr' (HELLAS-CH) MIS 5002735 (co-funded by the European Union and Hellenic National funds within the Operational Programme 'Competitiveness and Entrepreneurship'). 3D MHD simulations were performed with the use of the code PLUTO. The computations using the MHD code PLUTO were performed on 'ARIS' National HPC Infrastructure of the Greek Research and Technology Network (GRNET) under the project PluPS-2. We would also like to acknowledge Dr. N. Vlahakis for his support and suggestions on using Pluto.

## Author contributions

E.K. developed and performed the numerical simulations. V.D. was responsible and supervised the numerical work. Experiments were carried out by I.F. and A.S. G.K. contributed to the MHD simulations. N.A.P., M.B., E.L.C and I.K.N. gave crucial inputs to the experimental work as well as to the understanding of the physics. M.T. proposed the main idea of the study, supervised the experimental and numerical studies and was responsible for the research team.

## Additional information

**Competing interests:** The authors declare no competing financial interests.

