## [Peer Review File · Nature Communications]

Reviewers' comments:

Reviewer #1 (Remarks to the Author):

I have expressed all my views in the file attached.
According to my mind this manuscript could be published in this form (with potential several minor typos corrected at first).

In this manuscript the authors decided to have a closer look to an issue of the influence of the pre-plasma dynamics of a solid material to the generation and the dynamics of the plasma instabilities. This issue has been for long suspected by many as potentially rather significant. However, so far, due to its complexity, it has never been subject of any sufficiently detailed studies, neither experimentally nor theoretically, thus making difficult to get any clearer idea about how significant this issue could really be.

The authors of the submitted manuscript showed a considerable courage by taking upon themselves this intriguing task and performed both *theoretical* as well as *experimental* study of this effect in order to provide some insight to this very complex issue. They call their study quite rightly as a *proof of principle*. As they cleverly used just one particular scenario (experimental arrangement) which was accessible for this kind of research the most convenient way – the studied of the *Z-pinch*. Generalization of their results to some other scenarios, not accessible to such detailed studies, can be thus rather tentative. But the warning finger has been already risen by them.

It might be of interest making some comment to this issue of *generalization/validity* of their results to other experimental scenarios (e.g., laser-matter interaction) by the authors. And in this context also to an influence of laser pre-pulses.

The manuscript as well as the complimentary material provided to the reviewer was prepared very carefully and is written clearly enough to be valuable (an inspiration) to other interested researchers in this field.

I was able to find only very **few minor typos**:

In Legend 1:

Page 6, line 10:

...As time progresses, coronal plasma is generate, ...missing "d" in the word "generated"

Page 6, line 18:

...(8913 Kg/m³)... lower case „k" should be used instead The same can be found in the upper left part of the Fig. 1. **In the**

Supplement:

Fig. S.4, bottom right:

Intergrator... should read *Integrator*...

Reviewer #2 (Remarks to the Author):

This is an interesting and well-executed paper, and I am happy to recommend it for publication. It is fortuitous that the authors used a Z-pinch for experimental validation, as I believe the results may be of interest in the analysis of instabilities in magnetized liner fusion, as well as other imploding liner experiments.

My only suggestion relates to figure 1. The figure looks nice, but is fairly busy, and I found the labeling of sections a1-a4 somewhat unclear on a first reading. At the least, it would be helpful if it is reproduced at full size in the published article, as it would be difficult to read otherwise. Ideally, I would suggest breaking it into separate figures, but this should not be construed to mean that it is unsuitable for publication as is.

Reviewer #3 (Remarks to the Author):

The main purpose of the authors is to establish a connection between the processes that occur during the pre-plasma stages of Ohmically heated wires, and subsequent plasma instabilities (MHD instabilities in the current manuscript). The authors fail to demonstrate that due to the following main two reasons:

1. MHD instabilities refer to well defined solutions of the linearised MHD equations that represent exponentially growing perturbations with well defined spatial structures, namely well defined wave lengths. Each wave length in turn is uniquely characterised by a growth rate, a relationship that does not depend on the initial conditions. None of this is discussed and referred to in the manuscript and no relationship to known MHD instabilities is discussed.
2. The very purpose of finding the above mentioned connection is not that meaningful and cannot lead to my mind to deeper insight into the plasma dynamics. As mentioned in the previous item plasma instabilities emerge from random perturbations such that the structures with the largest growth rates dominate the others and information of the initial conditions is lost. So the initial perturbations are most probably indeed rooted in the pre-plasma stage but this is not really very important to the subsequent development of the plasma instabilities if they exist.

I recommend not to accept the manuscript for publication in Nature Communications.

Reply to reviewer #3

General communication with the reviewer:

We thank the reviewer for giving us the opportunity to discuss about certain aspects of our work.

Our manuscript presents work related to the study of the thermo-elasto-plastic phase of a metallic solid material as a seed for plasma instabilities. It describes in detail (main manuscript plus the supplementary material) a coupled thermo-elasto-plastic – electromagnetic – resistive MHD study which includes the intrinsic physical characteristics of the target. Furthermore, it takes into account all phases from solid to plasma. The multiphysics equations are solved dynamically using the Finite Element Method (FEM). This is the first time that such a study takes place. In particular, starting from the solid target the Johnson-Cook material model strength equations, for the study of the thermo-elasto-plastic phase of the solid material, are solved. The Burgess Equation of State (EOS) for the electrical resistivity (vs. temperature and density) is implemented in the study to accurately address the change in resistivity as a function of temperature and density. The temperature-dependent properties of the thermomechanical expansion of the material, its thermal conductivity and specific heat, as well as the latent heat of the melting phase have also been included in the study. The results of the spatiotemporal dynamics of the thermo-elasto-plastic phase of the material are then coupled to the resistive MHD equations to study the plasma formation. However, when a material is subjected to high energy deposition, impact or deformation at high strain-rate, it suffers large changes in its thermodynamic state. For this reason, a multi-phase EOS is needed, to describe all the possible states of matter. To incorporate this, a combination of the analytical Gruneisen and the tabular SESAME multiphase EOS have been used. All of the above establish a **new physics and methodology approach** for the study of the seed perturbations that trigger plasma instabilities taking into account the real physical and mechanical properties of the material. Furthermore, the manuscript presents an experimental study of the above physics starting from the thermo-elasto-plastic until the plasma regime, using a combination of laser probing diagnostics. For the early time dynamics of the solid material a modified Fraunhofer diffraction laser probing method of high spatial resolution ($\sim 1\mu\text{m}$) was implemented offering the opportunity to measure the thermo-elasto-plastic expansion of the target at very early times within the Joule heating. This offers an unambiguous comparison and feedback with the validated multiphysics approach providing an accurate physics platform for our study.

The main scope of our work at this stage was not to discuss about all aspects of plasma instabilities (kind of instability, growth rates, wavelengths etc.) since, within the plasma phase of matter, these are very well studied over the years, under certain conditions, for stationary and non-stationary systems. Our purpose was to study the physics, which takes place at the very first moments within the Joule heating including the intrinsic real physical properties of the target. **In other words, we do not start our study from an MHD plasma state at solid density but from a real solid.** This is a subject of great

importance for the last 15 years or so but also of high complexity as widely recognized by the scientific community (5-15, 19-20, 26-27, 31). The inclusion of the early plasma dynamics results in the manuscript only serves the necessity to demonstrate the validity of the new physics methodology presented via the comparison with experimental data. The presented physics and methodology of study opens new prospects/perspectives in the study of the seeding mechanisms of plasma instabilities because such a methodology a priori includes the real intrinsic physical characteristics of the material. Exactly this issue is of very high importance to a variety of plasma studies and applications, such as in pulsed-power liner inertial fusion studies, ICF fusion studies and in all plasma studies that incorporate the development of the Magneto-Rayleigh-Taylor (MRT) instability and its seeding processes.

Reply to the comments:

1. MHD instabilities refer to well defined solutions of the linearised MHD equations that represent exponentially growing perturbations with well defined spatial structures, namely well defined wave lengths. Each wave length in turn is uniquely characterised by a growth rate, a relationship that does not depend on the initial conditions. None of this is discussed and referred to in the manuscript and no relationship to known MHD instabilities is discussed.

2. The very purpose of finding the above mentioned connection is not that meaningful and cannot lead to my mind to deeper insight into the plasma dynamics. As mentioned in the previous item plasma instabilities emerge from random perturbations such that the structures with the largest growth rates dominate the others and information of the initial conditions is lost. So the initial perturbations are most probably indeed rooted in the pre-plasma stage but this is not really very important to the subsequent development of the plasma instabilities if they exist.

Comment 1: MHD instabilities refer to well defined solutions of the linearised MHD equations that represent exponentially growing perturbations with well defined spatial structures, namely well defined wave lengths. Each wave length in turn is uniquely characterised by a growth rate, a relationship that does not depend on the initial conditions. None of this is discussed and referred to in the manuscript and no relationship to known MHD instabilities is discussed.

Answer:

The Reviewer, as far as we can understand, refers to the linear MHD stability. However, a precise and complete study of MHD instabilities requires solving the full nonlinear, time-dependent MHD equations. This is a problem of high complexity both numerically and analytically. Due to this, the standard approach of approximation/simplification in the study of the ideal MHD stability problem has been the very well known linear stability treatment. This can be stated as the study of the stability of a specific MHD equilibrium state against small perturbations that do not interact with each other. The MHD equations are then linearized by expanding the solution around the equilibrium and by taking into account only the first order terms. Then, the initial and boundary conditions of the system of equations are included in order to solve an initial boundary value problem. Furthermore, assuming that the eigenmodes are orthonormal, the equations are reduced to ordinary differential equations. These are then treated by using the well known boundary value methods. The above

methodology determines all the eigenmodes (normal modes) of the system and their linear growth rates. However, the linear theory is a first order approximation that predicts the exponential growth or decay, or the oscillation of the initial perturbations (by solving the dispersion relation). An instability is present if ω is complex. Usually, the solutions are complex conjugate pairs, meaning that one of these will always be unstable, unless all of the roots are real. The growth rate is $\gamma = \text{Im}(\omega)$ that does not itself depend on the amplitude of the initial perturbations. However, the initial amplitude of the perturbations is important since different initial amplitudes will drive the system into different amplitudes of the developed instability at different times.

Let's consider the time-dependent part of an eigenmode i.e., $\psi_k = C_n e^{\gamma_k t}$ (for any mode k) having a growth rate γ_k which is independent on the C_n in the linear theory. Let's apply to this general solution two cases of the same mode k having different initial amplitudes C_1 and C_2 and requiring to have the same result let's leave it to evolve in time. The two solutions will reach equal values at different

moments. Performing the simple algebra we obtain that $\Delta t = \left| \ln \left(\frac{C_2}{C_1} \right) \right| \gamma_k^{-1}$. We can

easily see that the initial amplitudes C_1 and C_2 define when the mode will reach a certain amplitude value. Adding some extra simple algebra, we can reach at the same conclusion even if the more general case is considered where the instantaneous growth rate $\gamma_k(t')$ is taken into account.

For studies such as those described in our paper the growth rate γ_k^{-1} is \sim tens of

nanoseconds and because $\left| \ln \left(\frac{C_2}{C_1} \right) \right|$ is between 1 and 10 for most studies, Δt is of

that order. We should not forget that we are interested in stability where the growth time of the instability is long enough in order to achieve the goal of the plasma application under study (e.g., ICF studies, dense Z plasmas, high intensity laser-matter interactions etc.). In other words, we require stability with respect to perturbations that grow in time shorter than a characteristic time needed for the temporal plasma dynamics of the plasma of interest. A confusion can arise from the fact that randomly spatial distribution perturbations e.g. of density or temperature (or any other plasma parameter) are used as initial conditions. The Fourier transform of such distribution in the space domain leads to the initial values of the coefficients C_n of the general solution of the eigenvalue equation $-\rho_0 \omega^2 \xi = F(\xi)$, i.e. $\psi_k = C_n e^{\gamma_k t}$. Such trials with various pseudorandom functions lead to matching simulations and experiments for at least the fundamental wavenumbers (i.e. those with highest growth rates), and play the role of the "calibration" of the simulation or analytical study. Furthermore, since any linear superposition of the eigenfunctions is also a solution, the full match with experimental data in reality needs a full spectrum of initial perturbations.

Here, we should also explain the important difference between the stability studies of an imploding or exploding plasma dynamic system (non-stationary system) and the stability of a steady-state system (stationary system). For a non-stationary plasma the implosion or explosion takes place within a finite time, while in a steady-state plasma system lasts essentially forever. In both stationary and nonstationary systems, studying instabilities in the linear approximation an instantaneous increment $\gamma_k(t')$ defines the instantaneous

growth rate, which varies in time together with the unperturbed state of the system. To calculate the growth rate over a finite time period, the $\exp\left(\int_{t_1}^{t_2} \gamma_k(t') dt'\right) = \exp(\Gamma_k)$ should be defined. In non-stationary systems, such as the one in our study, the instabilities that have no time to grow are those for which $\Gamma \leq 1$. These are not destructive for the system. However, the instabilities for which $\Gamma \gg 1$ are destructive for the system. For such systems the growth rate of the instabilities is much higher than the life time of the plasma system as in most plasma applications of interest of the present study e.g., laser inertial fusion studies, magnetized liner inertial fusion studies (MagLIF), dense Z-pinch and X-pinch plasma studies and high intensity laser matter interactions. Therefore, the system is seriously destructed before the equilibrium preferable instability mode dominates [1, 2, 3, 4a,b, 23]. Thus, the mitigation of the poisonous role of the instability requires smaller initial perturbations (e.g. see review articles 5, 15). For a stationary system on the other hand, if for any reason some instability is present in the steady-state plasma geometry, the perturbations can reach at a nonlinear stage or at the preferable mode, independently from the initial perturbations. The saturated unstable plasma behavior can then exist for as long as the plasma does, provided that there is an external energy source to sustain it. We quote here some reviews of such plasma systems and geometries [1, 2, 35-39]. Moreover, and in addition to all the above, it is noticed that linear theory fails to deal with nonlinear phenomena, such as amplitude saturation or non-linear coupling between different modes or harmonic generation [40].

For most plasma instability studies using linear or even nonlinear MHD theory the most crucial parameter is to correctly choose the initial perturbations imposed either by a predefined perturbation, usually sinusoidal with fixed wavelength and amplitude, or by randomly seed perturbations as initial conditions [e.g., see 7, 15, 41 and references therein]. Ideally, numerical simulations should foresee the growth of instability from the natural physical parameters of the target material. This complex problem is broadly discussed over the last decade [e.g., see 7, 15 (review), 30 (review)]. Practically, in numerical simulations trial tests are performed using different functions and levels of initial perturbations until computational results match the experimental data. Using plasma parameter perturbations derived either from pseudorandom functions or from empirical laws can be valuable for studying instability growth rates if when “calibrated” against experimental results. This certainly cannot lead to a scientifically validated prediction ability, since it is very well known that different initial target conditions demand different initial conditions in order for simulations to match the experiments [e.g., see work in 48-50].

Contrary to this practice, our study incorporates the intrinsic real physical characteristics of the material as initial conditions: the initial perturbations which act as a seed for the growth of plasma instabilities are emerging as a result of the initial natural processes of the thermo-mechanical behavior of the heated material.

Complementary to the above analysis for the up-to-date prevailing physics on plasma instabilities, we present results (as many other authors have presented in the past) of our MHD simulation studies which show the influence, of the “externally” seeded into the system initial perturbations, for the plasma dynamics. Figs. 1 and 2 present plasma density contour plots for two different

times showing the spatial distribution dynamics of the plasma density. For comparison, Figs. 1(a) and 2(a) present results when the real thermo-elasto-plastic seed (as determined in our study) is used as initial perturbation. Images show different instability structure in both amplitude and wavelength. These results clearly and unambiguously show that the initial seed perturbations play important role to the non-stationary plasma system spatiotemporal dynamics.

Comment 2:

The reviewer also comments that “...no relation to known MHD instabilities is discussed”. And continues in comment 2: “The very purpose of finding the above mentioned connection is not that meaningful and cannot lead to my mind to deeper insight into the plasma dynamics. As mentioned in the previous item plasma instabilities emerge from random perturbations such that the structures with the largest growth rates dominate the others and information of the initial conditions is lost. So the initial perturbations are most probably indeed rooted in the pre-plasma stage but this is not really very important to the subsequent development of the plasma instabilities if they exist.”

Answer:

The authors, based purely on physics grounds, disagree with the opinion that “...the initial perturbations are most probably indeed rooted in the pre-plasma stage but this is not really very important to the subsequent development of the plasma instabilities if they exist.”, based purely on physics grounds. Furthermore, as per the “...no relation to known MHD instabilities is discussed”, as explained previously and will be repeated here, the goal of our study is not the dynamics of the instabilities in the plasma phase. These are very well studied over the years and are being continuously under investigation. Our research emphasizes on the initial thermo-elasto-plastic dynamics of the target acting as the necessary seed for the generation of plasma instabilities. Our proof-of-principle approach will be continued by specialization and extension to other target heating methods and geometries (e.g., heated targets by laser pulses or liners), where issues such as the nature of the instabilities or the growth rates, will be addressed. Such analytical studies are in progress.

Nevertheless, for the purpose of the present reply to the reviewer, an analysis about the concept of the electrothermal instability (ETI), which is the basis of the unstable behavior of the target in our study, will follow. **This instability is modified by the inclusion of the mechanical properties of the target, but this is a study to follow when our proof-of-principle approach will be specialized to various heating processes and target geometries.** A large part of the fundamental interest on the electrothermal instabilities during the last decade, among others lies in inertial confinement fusion studies, either in pulsed power liners or in ICF targets using lasers. For instance, surface roughness and defects of the targets is an issue of front-line research in the National Ignition Campaign since simulations show that they play an important role on the necessity for the mitigation of the MRT instability growth [13, 29, 30]. At the magnetized liner inertial fusion (MagLIF) concept it is of fundamental importance to understand the processes that occur during the target surface plasma initiation [45-47] as well as to study the influence of the surface

roughness of the liner targets on the MRT instability growth [6-7, 12-13, 44]. In this context, the electrothermal instability is particularly important because its k -vector allows for efficient coupling to the MRT instability growth.

When a material with temperature-dependent electrical resistivity is Ohmically heated at the skin effect mode (as in our study), it can be subject to the development of ETIs [1, 15-21, 26-29], where a temperature perturbation due to Joule heating can grow in time and space [1, 7-21, 26-30]. An ETI begins to grow immediately after the heating source is applied and persists as the outer part of the target expands rapidly. If $d\eta/dT > 0$ (where η is the electrical resistivity and T the temperature of the material) thermal instabilities give rise to layered structures called “strata”, which are developed normal to the current flow [19-27]. This type of instability dominates at current densities greater than 10^7 Acm^{-2} . Sometimes this instability is also referred to as “overheat instability” [15,32-33]. On the other hand when $d\eta/dT < 0$ as for example in the Spitzer resistivity of a plasma, ETIs give rise to filamentation of the current flow into channels [e.g. see 14-15 and references therein]. Furthermore, it is very well known that during an electrical explosion – for example due to Joule heating – the material undergoes all phases of matter from solid to plasma. **The ETI during the explosion of the target at the aforementioned or even larger current densities has become the subject of active research in the last 10-15 years or so due to its importance as a potential seed mechanism for the MRT instability growth.** The growing ETI gives rise to pressure nonuniformities at the surface acting as the potential seed for the MRT instability, which appears when the magnetic pressure increases up to the point able to oppose the rapid expansion of the material. At this point, the MRT instability – also known as flute instability – begins to grow seeded by the ETI of the previous phase. In the MRT instability, the plasma acts as a heavy liquid, the magnetic field that surrounds the conductor as a light liquid while the curvature of the magnetic force lines as the gravitational force. The growth rate of the MRT instability for an idealized system can be approximated as

$$\gamma_{MRT} \approx \frac{v_{Ti}}{(\rho R)^{1/2}} \left(\frac{\partial \rho}{\partial r} \right)^{1/2}$$

where v_{Ti} is the ion thermal velocity, R is radius of curvature of the magnetic field and ρ is the density [1, 4a,b]. The idealized models assume a stationary state where the magnetic pressure balances the thermal pressure. Dissipative processes such as magnetic field diffusion are not taken into account. But even so, analytical solutions cannot be calculated. The above equation for instance, gives the growth rate in the long wave limit [e.g. see 5 and references therein].

Other studies have pointed out the importance on the development of ETI of the nonlinear magnetic diffusion into the target during electrical explosions [20,42-43]. During an electrical explosion of a conductor at the skin effect mode the formation of a dense low temperature plasma at the surface of the conductor is accompanied by the generation and propagation of a shock wave and a nonlinear magnetic diffusion wave (NMDW). For the majority of metals the magnetic induction threshold above which a NMDW can take place is a few hundreds of kiloGauss. In this case the NMDW propagates in the conductor together with the shock wave generated at the surface. As a result of the heating of the metal and as time progresses, the magnetic diffusion rate increases due to

the increase of the electrical resistivity of the metal. It was shown that when a NMDW propagates through a conductor the long wavelength modes of the ETI are suppressed and that the short wave modes of ETI can be unstable [20, 51]. Recent studies with the inclusion of heat conduction, shock-wave propagation as well as viscous damping, also exposed the crucial role of predetermined ETI at the conductor surface to the growth of the MRT instability [20,42-43,51]. It was also proposed that in pulsed power magnetic liners the initial seed of the MRT instability growth is the electrothermal instability [6] and not the surface roughness. Experiments showed results on the mitigation of the growth of the MRT instability on metallic rods and liners when dielectric coatings were applied at the surface to mitigate the ETI seed [12-13].

Therefore, it is clear and widely recognized by the scientific community [e.g., see 15, 43-44 and references therein] that the complexity of such a multiphysics problem builds serious difficulty for the simulations on magnetically driven and laser ICF implosions, to successfully incorporate the realistic physical initial conditions of the targets. In addition, the use of assumptions on the plasma conditions after the target surface plasma has been initiated deteriorates the efforts towards the necessity to explore methods for mitigating the MRT instability. These uncertainties have been the trigger for the initiation over the last 15 years of dedicated investigation of the processes related to the initial target physical conditions. Up-to-date all studies were limited to the coupling of properties of the solid, among others the resistivity, the heat conduction, the magnetic diffusion, the viscous damping and the stress tensors, to the system of the MHD equations. These studies treat the target at its initial condition as an MHD fluid having a solid density, including terms to comprise some of the physical properties of the solid material.

Our study on the other hand, offers a new perspective to these efforts since it incorporates the physical initial material properties of the target in a electro-thermo-mechanical myltiphysics study of simulations and experiments. This is the major step in our study which we believe that alleviates some of the above difficulties and provides new perspectives in studying such plasma systems. Our study shows that the growth rates of the ETI that serves as the seed for the MRT instability **is at least one order of magnitude larger than those found in the literature so far.** This difference arises due to the treatment of the target as an electro-thermo-mechanical system, taking into account the target's mechanical and physical properties.

Summarising, due to the basic nature of the comments of reviewer #3, a detailed answer has been presented by the authors dealing with these comments as a whole, i.e. starting from the basic plasma instabilities issues studied over the years and reaching to the current demands for the better understanding of the seeding processes of plasma instabilities in plasma systems of significant importance today. Although this is a lengthy document, it was necessary to clarify the arguments raised by the reviewer in an unambiguous way and to demonstrate both the importance and scientific validity of our research. The authors strongly believe that the reviewer will reconsider his/her opinion about our submitted manuscript and we thank him/her for giving us the opportunity to discuss in more detail his/hers concerns.

References

- [1] Handbook on Plasma Instabilities, Volumes 1-3, Ferdinand F. Cap, Academic Press, New York, Volumes 1 (1976), Volumes 2 (1978), Volumes 3 (1982)
- [2] Physics of High-Density Z-Pinch Plasmas, Michael A. Liberman, John S. De Groot, Arthur Toor, Rick B. Spielman, Springer Verlag, New York Inc, 1999.
- [3] Glenn Bateman, MHD Instabilities (The MIT Press 1978).
- [4a,b] a): S. Ishimaru, Basic Principles of Plasma Physics, Benjamin (Benjamin, New York, 1973), b): F.F. Chen, Plasma Physics & Controlled Fusion Plenum Press, New York (1984),
- [5] D. Ryutov, M. Derzon, and M. Matzen, Rev. Mod. Phys. **72**, 167 (2000).
- [6] K. J. Peterson, D. B. Sinars, E. P. Yu, M. C. Herrmann, M. E. Cuneo, S. A. Slutz, I. C. Smith, B. W. Atherton, M. D. Knudson, and C. Nakhleh, Phys. Plasmas **19**, 092701 (2012),
- [7] K.J. Peterson et. al, Phys. Plasmas **20**, 056305 (2013)
- [8] V. I. Oreshkin, Tech. Phys. Lett. **35**, 36 (2009).
- [9] V. I. Oreshkin, R. Baksht, N. Ratakhin, A. Shishlov, K. Khishchenko, P. Levashov, and I. Beilis, Phys. Plasmas **11**, 4771 (2004).
- [10] V. I. Oreshkin, Phys. Plasmas **15**, 092103 (2008).
- [11] A. G. Roussikh, V. I. Oreshkin, S. A. Chaikovsky et al., Phys. Plasmas **15**, 102706 (2008).
- [12] K.J. Peterson, T.J. Awe, E.P. Yu et al., Phys. Rev. Lett. **112**, 135002 (2014).
- [13] T.J. Awe, K.J. Peterson, E.P. Yu, et al., Phys. Rev. Lett. **116**, 065001 (2016).
- [14] M.G. Haines, Phys. Rev. Lett. **47**, 917 (1981).
- [15] M.G. Haines, A review of Dense Z-pinch, Plasma Phys. Control. Fusion **53**, 093001 (2011)
- [16] K. S. Fansler and D. D. Shear, in Exploding Wires, edited by W. G. Chace and H. K. Moor (Plenum, New York, 1965), p. 185.
- [17] . A. Valuev, I. Ya. Dikhter, and V. A. Zeigarnik, Zh. Tekh. Fiz. **48**, 2088 (1978).
- [18] Y. G. Epelbaum, Zh. Tekh. Fiz. **54**, 492 (1984).
- [19] V. I. Oreshkin, R. B. Baksht, N. A. Ratakhin et al., Phys. Plasmas **11**, 4771 (2004).

- [20] V. I. Oreshkin and S. A. Chaikovsky, *Physics of Plasmas* **19**, 022706 (2012)
- [21] B. B. Kadomtsev, in *Reviews of Plasma Physics*, edited by M. A. Leontovich (Consultants Bureau, New York, 1980).
- [22] S. I. Braginsky, in *Reviews of Plasma Physics*, edited by M. A. Leontovich (Consultants Bureau, New York, 1980).
- [23] Yu. P. Raizer, *Gas Discharge Physics* (Springer, Berlin, 1997).
- [24] D. B. Sinars, M. Hu, K. M. Chandler, T. A. Shelkovenko, S. A. Pikuz, J.B. Greenly, D. A. Hammer, and B. R. Kusse, *Phys. Plasmas* **8**, 216 (2001).
- [25] G. S. Sarkisov, P. V. Sasorov, K. W. Struve, and D. H. McDaniel, *J. Appl. Phys.* **93**, 1674 (2004).
- [26] A. G. Rousskikh, V. I. Oreshkin, S. A. Chaikovsky, N. A. Labetskaya, A.V. Shishlov, I. I. Beilis, and R. B. Baksht, *Phys. Plasmas* **15**, 102706 (2008).
- [27] V. I. Oreshkin, A. G. Rousskikh, S. A. Chaikovsky, and E. V. Oreshkin, *Phys. Plasmas* **17**, 072703 (2010).
- [28] S. I. Anisimov and Ya. B. Zeldovich, *Pis'ma Zh. Tekh. Fiz.* **3**, 1081 (1977).
- [29] S.W. Haan, J.D. Lindl, D.A. Callahan et al., *Phys. Plasmas* **18**, 051001 (2011)
- [30] R. S. Craxton, K. S. Anderson, T. R. Boehly et al., *Physics of Plasmas* **22**, 110501 (2015)
- [31] J. D. Pecover and J.P. Chittenden, *Phys. Plasmas* **22**, 102701 (2015)
- [32] V. I. Oreshkin, *Tech.Phys.Lett.* **35**,36 (2009)
- [33] M.G Haines, *J. Plasma Phys.* **12**, 1 (1974)
- [34] B.B. Kadomtsev, *Reviews of Plasma Physics*, **vol2.** ed. M.A. Leontovich, p.153, NewYork: Consultants Bureau.
- [35] M.G. Haines, *Phil. Trans. R. Soc. Lond*, A300649 (1981)
- [36] M.G. Haines, *Astrophys.SpaceSci.* **256**, 1 (1998)
- [37] A.E. Dangor , *PlasmaPhys.Control.Fusion* **28**, 1931 (1986)
- [38] N.R. Pereira and J. Davis, *J. Appl. Phys.* **64** R1–27 (1988)
- [39] M.K. Matzen, *Phys.Plasmas* **4**, 1519 (1997)
- [40] F.A. Cap, *Waves and Instabilities in Plasmas*, Springer-Verlag (1994)
- [41] Paraschiv et al., *Phys. Plasmas* **17**, 072107 (2010)].
- [42] S.A. Chaikovsky, V. I. Oreshkin et al., *Phys. Plasmas* **21**, 042706 (2014)

- [43] S.A. Chaikovsky, V. I. Oreshkin et al., Phys. Plasmas **22**, 112704 (2015)
- [44] T.J. Awe et al., IEEE Transactions on Plasma Science **45**, 584 (2017)
- [45] I. Lindemuth, R. Siemon, B. Bauer et al., Phys. Rev. Lett. **105**, 195004 (2010).
- [46] T. J. Awe, B. S. Bauer, S. Fuelling, and R. E. Siemon, Phys. Rev. Lett. **104**, 035001 (2010).
- [47] S. F. Garanin, S. D. Kuznetsov, W. L. Atchison et al., IEEE Trans. Plasma Sci. **38**, 1815 (2010)
- [48] D. L. Peterson, R. L. Bowers, J. H. Brownell et al., Phys. Plasmas **3**, 368 (1996).
- [49] D. L. Peterson, R. L. Bowers, K. D. McLenithan et al., Phys. Plasmas **5**, 3302 (1998).
- [50] D. L. Peterson, R. L. Bowers, W. Matuska et al., Phys. Plasmas **6**, 2178 (1999).
- [51] V. I. Oreshkin S.A. Chaikovsky, et al., Phys. Plasmas **23**, 122107 (2015)

Figure 1

Figure 2

Fig.1&2: Plasma density contours at 210ns and 240ns from the current start respectively. (a): initial perturbation seed is produced by the real thermo-elasto-plastic dynamics of the material (our study), (b,c,d): the seed is generated by an artificial sinusoidal function with 1.5 larger (b), equal (c) and 1.5 smaller (d) wavelength but same amplitude as in (a), are used respectively as initial seed. (e, f, g): random seed perturbations functions (e, f) and a multispectrum periodic function (g) are used as initial seed respectively. Indicatively the function in (g) is:
 $\cos(x)+0.5*\cos(3*x+23)+0.5*\cos(5*x-0.4)+0.5*\cos(7*x+2.09)+0.5*\cos(9*x-3)$ and is plotted in Figure 3.

Figure 3: plotted function: $\cos(x)+0.5*\cos(3*x+23)+0.5*\cos(5*x-0.4)+0.5*\cos(7*x+2.09)+0.5*\cos(9*x-3)$ which serves as initial density perturbation in Figure 1(g) & 2(g).

Reviewers' comments:

Reviewer #2 (Remarks to the Author):

I have reviewed the authors' rebuttal to reviewer number 3, and I believe the authors have very satisfactorily addressed the points raised. I recommend the paper for publication.

Reviewer #3 (Remarks to the Author):

I do appreciate the amount of the numerical as well as experimental work invested in carrying out the research, but I am still of the opinion that the manuscript is at best premature. Instability is the main motif of the manuscript, repeatedly referred to, and claimed by the authors to appear at all stages of evolution, from solid to plasma state. Yet, none of the well known instabilities is referred to or identified, there is no discussion whatsoever of the physical nature of the instabilities, and no connection is being made with any instability theory, linear or nonlinear. For instance, claiming to observe an instability at each stage of the evolution, the authors do not demonstrate that the physical conditions are indeed commensurate with theoretical criteria for the onset of some particular instability, as it is reasonable to expect that at each stage the instability (if exists) is of different physical origin. Unfortunately therefore, I see no compelling reason to change my original recommendation against publication.

Reviewer #4 (Remarks to the Author):

I was asked by the Editor to comment "on the relevance and importance of the issues raised by Reviewer #3" and to provide an "independent assessment of the work as a whole".

Reviewer #3 had two Comments. The first Comment concerns the complete lack of discussions on the unstable eigenmode solution of the MHD instabilities. Reviewer #3 made the valid point that these eigenmode solutions are independent of the initial conditions, which would make the relevance of this paper questionable as the present manuscript is restricted only to the study of initial conditions. I wish to point out that the "initial conditions" that the authors considered occurred long before the MHD instabilities come into play. There remains a big gap in our understanding of how the initial conditions of the sort considered by the authors would evolve into the seed for the global, potentially destructive MHD instabilities. I do not feel that the latter problem needs to be resolved in this paper. The authors made a real contribution by identifying the need to comprehensively account for the real physics of a solid target, its thermo-elasticplastic response to an intense drive current. There is considerable experimental evidence that such initial, material-dependent surface instability would somehow seed the global MHD mode(s) that become important at a later time. Even though the authors did not address such a process, the merit of this paper is hardly diminished. I would suggest the authors make a statement, something like, "How would the instabilities studied in this paper evolve into the seed of a global MHD mode at a later time remains an open question."

The second Comment by Reviewer #3 is questionable in its merit. Reviewer #3 dismissed the importance of the initial conditions for the eventual development of the instabilities. The authors gave a convincing rebuttal of Review #3 on this. I may add that while it is unclear what seeds the MHD instabilities when these instabilities first appear, but the mode that is seeded may turn into the preferred mode even though that mode is not necessarily the most unstable eigenmode in the subsequent evolving “equilibrium”. Thus, seeding (sometimes called “priming” in other areas) could be much more important than the intrinsic instability growth in the study of mode excitation. The recent experiments by Yager-Elorriaga et al. [A], [B] on magnetized imploding thin foils showed strong evidence of the importance of seeding, and the persistence of the initially seeded mode from implosion, stagnation, and explosion. These closely related papers should perhaps be cited,

[A] D. A.Yager-Elorriaga, et al., “Seeded and unseeded helical modes in magnetized, nonimploding cylindrical liner-plasmas,” Phys. Plasmas 23, 101205 (2016).

[B] D. A.Yager-Elorriaga, et al., “Discrete helical modes in imploding and exploding cylindrical, magnetized liners,” Phys. Plasmas 23, 124502 (2016).

It is my judgment that the authors have made a valuable contribution in their dedicated study of the possible initial conditions that eventually seed the macroscopic, global MHD instability mode.

Finally, I am curious about the authors' response to the following three (3) questions:

1. For the electro-thermal-mechanical (ETM) instability that they studied, do you have any new insight into the mechanism that converts the $\sim 1\mu\text{m}$ axial wavelength in the surface perturbations to the ~ 100 to $300\mu\text{m}$ axial wavelengths that usually appear at a later time?
2. In your study of ETM, did you determine k , the wave vector of the perturbations, and its evolution in time?
3. If you apply a small, external axial magnetic field, would you (expect to) see helical structures in ETM? If so, what is m , the azimuthal mode number?

Reply to reviewer #4

The authors would like to thank the reviewer for his/her positive response as well as for his/her valuable comments and suggestions. The questions raised by the reviewer indicate deep knowledge of the field of our work.

General communication with the reviewer

The seeding mechanisms of helical structures in premagnetised liner experiments, as those studied in ref 1-6 is a very critical issue when liner's stability is studied (or explored). The observed 3D helical structures persist throughout the implosion even though the azimuthal magnetic field greatly exceeds the imposed axial magnetic field at the liner's outer wall. Such structures (instabilities) contrast with the azimuthally correlated MRT instabilities that are consistently observed in non-premagnetised experiments. 3D MHD simulations (i.e. ref 5) showed that if a small amplitude helical perturbation is seeded on the liner's outer surface, it grows in amplitude and persists with almost constant pitch as the liner implodes even if $B_\theta \gg B_z$. As discussed in ref 3, reproduction of the experimentally observed helical structures from simulations without artificial seeding is very challenging. 3D MHD simulations up to date require helical initial seeding to reproduce the experimentally observed structures. The standard electrothermal instability (ETI) which is characterised by a \mathbf{k} vector perpendicular to the azimuthal \mathbf{B} (or \mathbf{k} parallel to the axis of the wire) when $d\eta/dT > 0$, i.e. in the condensed matter state of the material (η is the resistivity and T the temperature) needs the support from further studies, theories and methods in order to explain structures such as the helical structures in the presence of an axial magnetic field.

Our study indicates that thermomechanical perturbations (modes) are generated in the thermoelastic phase (solid) of the interaction of the solid target with the heating current, and it is quite probable that these modes are coupled and excited into helical structures when an external axial magnetic field premagnetises the target. In the light of the above the proposed electro-thermo-mechanical (ETM) instability can enlighten the study of seeding mechanisms of helical structures early in the discharge but further work is required to analyze in details its role and its importance.

Revision of the manuscript

We have revised the manuscript according to the comments and the questions raised. In particular we have added the following sentence in the discussion section of the paper "*How would the instabilities studied here evolve into the seed of a global MHD mode at a later time in the plasma phase remains an open question and requires further study*". Also, we have added a short paragraph at the end of the manuscript (as further study of our work) mentioning the recent studies on the observed helical structures in magnetised cylindrical liner plasmas in relation to the possible extension of our work.

The suggested papers [1-2] have been cited as well as the following relevant paper by M.R. Weis et al. [3] which is related to kink-like perturbations in liners (non-axisymmetric) and to the coupling of the $m=0$ MRT and sausage modes and $m=1$ MRT and kink modes.

Furthermore, the papers by T.J. Awe et al [4-5] and by M.R. Gomez et al. [6] have been cited since they are also related to the discussion about the helical kink like structures observed in the presence of axial magnetic fields in axially pre-magnetised ($B_z > 0$) MagLIF relevant liners.

References

- [1] D. A. Yager-Elorriaga, et al., Phys. Plasmas 23, 101205 (2016)
- [2] D. A. Yager-Elorriaga, et al., Phys. Plasmas 23, 124502 (2016)
- [3] M. R. Weis, et al., Phys. Plasmas 22, 032706 (2015)
- [4] T. J. Awe et al., Phys. Rev. Lett, 111, 235005 (2013)
- [5] T. J. Awe et al., Phys. Plasmas 21, 056303 (2014)
- [6] M.R. Gomez et al., Phys. Rev. Lett. 113, 155003 (2014)

Questions raised by the reviewer:

1. For the electro-thermal-mechanical (ETM) instability that they studied, do you have any new insight into the mechanism that converts the $\sim 1 \mu\text{m}$ axial wavelength in the surface perturbations to the ~ 100 to $300 \mu\text{m}$ axial wavelengths that usually appear at a later time?
2. In your study of ETM, did you determine \mathbf{k} , the wave vector of the perturbations, and its evolution in time?
3. If you apply a small, external axial magnetic field, would you (expect to) see helical structures in ETM? If so, what is m , the azimuthal mode number?

Answers:

Question 1:

The mechanism that converts the short wavelengths into long wavelengths is generally an open question and under investigation. The current view on this matter is that the shorter (axial) wavelengths merge into longer wavelength structures. The merging mechanisms are under study by various groups. Moreover, if smaller axial wavelength structures merge into larger axial wavelengths, a high azimuthal mode could also convert into lower modes. This may also explain the conversion of high azimuthal modes found in the thermoelastic phase of the target into lower azimuthal modes and finally the dominant sausage mode later in the plasma phase when no external axial magnetic field is present (please see below the related discussion regarding question 2).

Question 2

This question is related to the kind of the instability observed in our study. The displacement as a function of z and the azimuthal angle θ are shown in the figures below. The resulting wave vector \mathbf{k} has both k_z and k_θ components.

Figure 1, below, shows the displacement as a function of the azimuthal angle θ at various z-planes of the wire (ranging from 1.5 mm to 6 mm) at different times.

Figure 1 (a-e)

Figure 1. Displacement as a function of the azimuthal angle θ at various z-planes of the wire (six different z-planes) at different times

Also, Figure 2 below shows the displacement as a function of z at three different azimuthal angles (0, 45 and 90 degrees) at different times.

Figure 2. Displacement as a function of z at three different azimuthal angles (0, 45 and 90 degrees) at different times

It can be seen that both displacements do not consist of a single wavelength (for instance at least two and maybe three wavelengths are present at early times). In particular, the displacement as a function of the azimuthal angle θ indicates the existence of a spectrum of wavelengths. The dominant wavelength ($\pi/2$) seems to prevail over the others as time progresses from the thermoelastic to the melting phase of the material. When the phase changes into plasma the $m=0$ mode reveals as expected and predicted by the experiment. Similarly, the z dynamics show that the short wavelength structures merge into longer ones. As discussed above, if shorter axial wavelength structures merge into larger axial wavelengths, a high azimuthal mode could also convert into lower modes.

In Figure 3 the time evolution of the dominants k_z (at $\theta=0$ degrees) and k_θ (at $z=1.5\text{mm}$) in the thermoelastic phase of the material is also presented. Initially, the k_z is smaller than the k_θ but as the k_z decreases, the k_θ decreases faster until they become almost equal at the end of the thermoelastic phase. As discussed above this interesting observation is related to the conversion of small wavelengths into longer ones. These problems shall be left for future work.

Figure 3. Time evolution of the k_z and k_θ

Question 3

The question is associated to the recent studies on z-pinch liners pre-magnetised with an external axial magnetic field [e.g. ref 1-6]. The inclusion (both in experiments and in the FEM simulations) of an axial B-field in our study is an ongoing research topic as already mentioned.

We strongly believe (as evidenced by the spectrum of modes discussed above and the k_z and k_θ evolution) that the axial B-field will lead to helical structures as its presence may correlate the axial and azimuthal modes into helix-like perturbations but this is to be validated by further studies.

REVIEWERS' COMMENTS:

Reviewer #4 (Remarks to the Author):

I am satisfied with the authors' response. The revised paper is recommended for publication, as is.